# Production of Extracellular Matrix Proteins in the Cytoplasm of *E. coli*: Making Giants in Tiny Factories

**DOI:** 10.3390/ijms21030688

**Published:** 2020-01-21

**Authors:** Anil A. Sohail, Madhuri Gaikwad, Prakash Khadka, Mirva J. Saaranen, Lloyd W. Ruddock

**Affiliations:** Faculty of Biochemistry and Molecular Medicine, University of Oulu, 90220 Oulu, Finland; anil.sohail@oulu.fi (A.A.S.); madhuri.gaikwad@oulu.fi (M.G.); pkbiochemist99@gmail.com (P.K.); mirva.saaranen@oulu.fi (M.J.S.)

**Keywords:** protein production, *E. coli*, disulfide bonds, CyDisCo, extracellular matrix

## Abstract

*Escherichia coli* is the most widely used protein production host in academia and a major host for industrial protein production. However, recombinant production of eukaryotic proteins in prokaryotes has challenges. One of these is post-translational modifications, including native disulfide bond formation. Proteins containing disulfide bonds have traditionally been made by targeting to the periplasm or by in vitro refolding of proteins made as inclusion bodies. More recently, systems for the production of disulfide-containing proteins in the cytoplasm have been introduced. However, it is unclear if these systems have the capacity for the production of disulfide-rich eukaryotic proteins. To address this question, we tested the capacity of one such system to produce domain constructs, containing up to 44 disulfide bonds, of the mammalian extracellular matrix proteins mucin 2, alpha tectorin, and perlecan. All were successfully produced with purified yields up to 6.5 mg/L. The proteins were further analyzed using a variety of biophysical techniques including circular dichroism spectrometry, thermal stability assay, and mass spectrometry. These analyses indicated that the purified proteins are most likely correctly folded to their native state. This greatly extends the use of *E. coli* for the production of eukaryotic proteins for structural and functional studies.

## 1. Introduction

Protein production in both natural and recombinant systems can be broadly divided into two processes: synthesis of the polypeptide chain(s), and then folding to the functional, native conformation. For a significant proportion of human proteins, folding to the native state includes disulfide bond formation. In natural systems, oxidative folding, where folding is linked to the oxidation of dithiols to disulfides, takes place in specialized compartments [1]. In the case of Gram negative bacteria, this occurs in the periplasm [2], while in eukaryotes proteins are folded in either the endoplasmic reticulum (ER) [3] or mitochondrial intermembrane space [4].

Disulfide bond formation is a complex process, but it can be split into two steps, *de novo* disulfide bond formation i.e., oxidation of dithiols to the disulfide state, and subsequent isomerization of non-native disulfides. All compartments in which native disulfide bond formation occurs have catalysts of both steps. For example, the *de novo* reaction in the *E. coli* periplasm is catalyzed by DsbA/DsbB shuttle system while in eukaryotes it is catalyzed by enzymes from the sulfhydryl oxidase family e.g., Ero1 and Erv1p [5,6,7,8]. Similarly, the isomerization reaction in the *E. coli* periplasm is catalyzed by DsbC with the aid of DsbD, while in the ER of eukaryotes it is catalyzed by members of the protein disulfide isomerase (PDI) family [5,7,8,9].

For decades, *E. coli* was considered to be one of the best recombinant protein production systems as it is inexpensive, quick, scalable and it is easy to alter genetically [10,11]. However, one of the limitations of *E. coli* is the formation of post-translational modifications (PTMs), including the formation of disulfide bonds [7,12]. The cytoplasm of *E. coli* has a reducing environment, preventing native disulfide formation, while the periplasm has a smaller volume [13] and may have limitations connected with the capacity of the translocation from the cytoplasm [14], which combined mean that periplasmic yields are usually significantly lower than cytoplasmic expression yields. To help to overcome the limitations of native disulfide bond formation in *E. coli*, we used synthetic biology to introduce a system known as CyDisCo (cytoplasmic disulfide bond formation in *E. coli*) [15,16,17,18]. This system is based on heterologous co- or pre-expression of a eukaryotic catalyst of *de novo* disulfide bond formation (e.g., a sulfhydryl oxidase, most usually yeast Erv1p) as well as a eukaryotic catalyst of disulfide bond isomerization (e.g., a disulfide isomerase, most usually human PDI) i.e., catalysts of the two steps in native disulfide bond formation. Together these boost productive oxidative folding and hence the yield of disulfide-containing proteins [15,16,17,18].

A wide variety of prokaryotic and eukaryotic proteins containing disulfide bonds have been successfully produced using this system. However, most proteins reported to date as being produced have between one and five disulfide bonds (reviewed in [19]). The most complex protein reported to date was resistin, a homodimer with five disulfides in each monomer plus an inter-molecular disulfide [19]. This level of disulfide complexity is far below the level which can be produced in eukaryotic systems, with some extracellular mammalian proteins having in excess of one hundred disulfide bonds.

In this study, we examine the potential limitations of the CyDisCo system for native disulfide bond formation by expression of domain constructs of mammalian extracellular matrix (ECM) proteins containing between 8 and 44 disulfide bonds. No upper limit for disulfide production was found. All six constructs could be made soluble, and were purified in yields of up to 6.5 mg/L. Biophysical analysis by circular dichroism, thermal stability and mass spectrometry suggested that all six were present in a folded state and contained no free thiol groups. Hence, they are probably natively folded. To our knowledge, these include the most complex disulfide bonded proteins reported to be successfully produced in a prokaryotic system. This study extends the possibility of the use of prokaryotic systems for the production of eukaryotic proteins for structural and functional studies.

## 2. Results and Discussion

### 2.1. Disulfide-rich ECM Proteins as Model Proteins

CyDisCo, either as a single polycistronic plasmid-based system or more commonly as a dual plasmid-based system, has been used to successfully produce a range of eukaryotic proteins having typically between one and five disulfide bonds (reviewed in [19]). The successful production of these correctly folded proteins with the help of co- or pre-expression of Erv1p and PDI in the cytoplasm of *E. coli*, made us consider what the limitations might be in terms of the production of more complex disulfide bonded proteins. We hypothesized that it may be possible to make eukaryotic proteins with a higher number of disulfide bonds using the system, but that the yield and/or the homogeneity of the protein produced might reduce as the complexity increases.

ECM proteins are rich in disulfide bonds and hence they make good candidates to test the potential limitations of CyDisCo. We initially chose as our test proteins two ECM proteins containing von Willebrand Factor D (VWF-D) domains. Proteins containing these domains have associated disease states. For example, various VWF-D domain-containing proteins like BMP-binding endothelial regulator protein (BMPER), alpha tectorin and von Willebrand factor are associated with disease states such as diaphanospondylodysostosis, deafness autosomal recessive disorder and hemorrhagic disorders, respectively [20,21,22]. As such VWF-D domain-containing proteins are attractive subjects for structure-function relationship studies. Currently there is very limited structural information available for these domains. One reason for this may be the difficulty of producing these disulfide-rich proteins in a homogeneously folded state in good yields. To our knowledge they have previously only been produced in eukaryotic systems [23].

When this project started there were no structures available for a VWF-D domain from any protein. Very recently, the structure of one of the VWF-D domains from human VWF has been solved [23]. However, there are still numerous proteins containing a homologous domain whose structure is unknown. These include mucin 2 and alpha tectorin.

Mucin 2 is part of the mucus layer that lines multiple internal organs protecting them from both physical and chemical damage [24]. It also provides the first line of defense of the gastrointestinal tract [25], with the inner layer of the mucin 2-dependent mucus layers being devoid of bacteria [26,27]. The mucus layer is also an interesting study subject from the pharmacological point of view, as drugs need to interact with and penetrate the layer in order to reach their targets [24]. Alpha tectorin on the other hand is present in the tectorial membrane of the ear canal and plays a significant role in the propagation of sound [19]. Even though these proteins have different functions both of them contain four VWF-D domains. These domains are accompanied by a cysteine-rich C-8 region that is often followed by a trypsin inhibitor like (TIL) domain and an E-8 region. This entire assembly has recently been denoted as the D-assembly [23]. In mucin 2 three of the four D-assemblies are situated at the N-terminal of the protein while one is situated at the C-terminal (Figure 1A). In alpha tectorin the four D-assemblies are consecutive in the primary sequence and make up most of the protein (Figure 1B).

Since the chosen proteins of interest (POI) were 237 and 538 kDa in size and contain 146 and 214 cysteines respectively, we chose to make smaller domain constructs. Domain boundaries were based on sequence alignment and conservation between domains and between species, available literature and secondary structure prediction. Since this is prone to error, especially where no structure exists for any such domain and since domain-domain interactions may be essential to attain the native state, a number of VWF-D domain-containing constructs of mucin 2 and alpha tectorin with different putative domain boundaries were tested for soluble expression. Since expression can be dependent on the strain and expression conditions, up to four different *E. coli* strains, two media and two different expression temperatures were screened. For mucin 2 the (V36-G389) construct having the whole D assembly (VWF-D + C8 + TIL + E8) showed good results and was used for further studies. Similarly, for alpha tectorin a construct having VWF-D domain and C8 region (P701-P981) was taken forward. For both constructs the best expression condition was found to be at 15 °C in BL21(DE3) cells in rich autoinduction media.

The alpha tectorin (P707-P981) construct, with 283 amino acids (8 disulfide bonds) lies at the top end of proteins previously reported to be produced using CyDisCo [19]. In contrast, the cysteine-rich mucin 2 (V36-G389) construct which contains 15 disulfide bonds in the native state, both contains more disulfide bonds than previously reported to have been made in any protein using CyDisCo and contains a higher disulfide density (8.5% of amino acids are Cys found in disulfide bonds). Both of the constructs were successfully made and purified as soluble proteins using the CyDisCo system and had a single observable redox state in the purified state (Figure 2A). Purified yields were good, with 6.5 mg/L for the mucin 2 construct and 5.5 mg/L for the alpha tectorin construct produced in shake flasks in rich autoinduction media.

While the alpha tectorin construct behaved as all previously reported proteins produced using CyDisCo, the mucin 2 construct did not. Instead, after the IMAC and AnEx purification steps, it was observed that the mucin 2 construct was not present in a homogenous redox state (Figure 2B). Specifically, the non-reduced NEM treated samples showed separate bands at different molecular weights, suggesting that a fraction of the POI, or a degradation product of the POI, existed in a non-native redox state. However, during the subsequent purification using anion exchange and gel filtration chromatography we successfully isolated the natively folded protein (Figure 2B). Non-native redox states were not previously reported for other POI produced using CyDisCo. This could be an indication that folding to the native redox state may be a limiting factor for CyDisCo with more disulfide-rich POI. This effect was not observed for the alpha tectorin construct which contains only 8 disulfide bonds compared to the 15 in the mucin 2 construct.

### 2.2. Perlecan as a Model Protein

To further examine this and to test the possible limits of CyDisCo, we decided to express constructs of mouse perlecan. Perlecan is multi-functional basement membrane protein, which plays a vital role in the development of skeletal and cardiovascular muscles [28] and in strengthening of the mechanical and morphological properties of bone [29]. Perlecan is a large multi-domain protein, which has a cleavable signal sequence and no transmembrane regions. The mature protein is divided into five regions [30] (Figure 3A). These regions range in size from approximately 110 to 1300 amino acids. Perlecan contains 172 cysteine residues, with the highest density being in region 3. Mutations in region 3 of perlecan linked to Schwartz–Jampel Syndrome (SJS) and to dyssegmental dysplasia, Silverman–Handmaker type (DDHS) [31]. Region 3 is a combination of cysteine-rich regions and cysteine-free globular regions. There are three laminin IV type-A like domains which lack cysteine residues and eleven laminin Epidermal Growth Factor (EGF)-like domains which have 88 cysteine residues, all of which are thought to be in disulfide bonds. Three of the laminin EGF-like domains are split into two parts with the insertion of a laminin IV type-A like domain in between them. This splitting and insertion of units may have arisen during the evolution of perlecan due to alternative splicing events [32]. Another interesting feature of this region is that laminin EGF-like domain 4 is truncated in comparison to other EGF-like domains within the region. Additionally, there is a small unassigned sequence of 16 amino acids (G503-P520) between the Ig-like C2-type domain 1 from region 2 of perlecan and laminin EGF-like domain 1, which may be an essential part of this region [28,33,34,35,36]. In total region 3 is 1169 amino acids in length and is thought to contain 44 disulfide bonds. As such region 3 of perlecan would be challenging for any recombinant protein production system.

Initially, region 3 was divided based on putative domain boundaries into three smaller constructs (G503-Q874, S762-R1158, and R1158-T1672), which contain 26, 30 and 48 cysteines respectively. As such all three contain more disulfide bonds than any protein previously reported to have been made using CyDisCo. Despite the disulfide complexity, all three were successfully produced and purified as soluble proteins in *E. coli* using the CyDisCo system (Figure 3B). Furthermore, redox heterogeneity was not observed. Purified yields of 1.0 mg/L, 2.9 mg/L and 4.4 mg/L were obtained for G503-Q874, S762-R1158, and R1158-T1672 respectively from chemically defined autoinduction media.

These results were encouraging as, to our knowledge, such complex disulfide-rich proteins had not previously been reported in any *E. coli* expression system. To further test the potential limitation of CyDisCo, we attempted to express the whole of perlecan region 3 (G503-T1672), which is 126 kDa in size and is thought to contain 44 disulfide bonds in the native state. Despite the disulfide complexity of this construct it was also successfully produced in a soluble state (Figure 3B), with a purified yield of 0.9 mg/L from chemically defined autoinduction media and 4.4 mg/L from rich autoinduction media. Apart from yield no differences were observed in the protein produced in different media. All of the purified perlecan constructs ran at their expected molecular weight in SDS-PAGE and ran as a single band on non-reducing SDS-PAGE suggesting they had a single redox state.

Analysis by electrospray ionization mass spectrometry (ESI-MS) confirmed that the purified perlecan constructs had the expected molecular weight (Table 1) consistent with the constructs having all of their cysteines in disulfide bonds. The presence of any free, exposed cysteines in the constructs was further evaluated by treating selected samples with *N*-ethylmaleimide (NEM) prior mass spectrometric analysis. NEM-trapped samples would be expected to show an addition of 125 Da in the molecular weight of the protein for each free cysteine. None of the samples analyzed showed any increase in the mass after NEM-treatment, implying that none contained free thiols i.e., all cysteines were in disulfide bonds. Alkylation with NEM or iodoacetamide has a long history of use in studying in vitro oxidative refolding [37,38,39,40].

To ensure that the method was working correctly two controls were used. Firstly, mouse FGF7 made using CyDisCo for another project was subjected to the same treatment. This protein contains 5 cysteines and hence must contain at least one free thiol group. The theoretical average molecular weight of the mature protein (minus the initiating methionine, which should be removed by *E. coli* methionine aminopeptidase) is 19,713 Da. The experimentally determined mass was 19,707 Da in the absence of NEM and 19,834 Da after NEM treatment. This implies the protein has two disulfide bonds, plus one free thiol group. Secondly, reduced BPTI used as a substrate to study oxidative refolding [39,40,42,43] showed the expected mass with no NEM treatment and an increase in mass of 750 Da upon treatment with NEM, consistent with it having six free thiol groups corresponding to its six cysteine residues. These results confirm that the methodology works correctly and implies that all cysteines in the perlecan constructs tested are in disulfide bonds. There are reports of some proteins obtaining a native disulfide content during refolding in vitro while having non-native disulfide bonds, Hirudin being the best-characterized example [44]. However, this is not thought to be the norm and we are not aware of reports of an equivalent situation being reported during folding *in vivo*. Hence, our results suggest that native disulfide bond formation has been obtained for the perlecan constructs. To confirm a native disulfide bonds pattern has been obtained disulfide mapping is often undertaken. However, disulfide mapping of the purified proteins was not undertaken for two reasons. Firstly, the native disulfides have not been previously mapped, they are only presumed by similarity with proteins with homologous domains. Secondly, they are disulfide-rich with many spatially adjacent cysteine residues making mapping problematic e.g., 73 of the 88 Cys found in perlecan region 3 have spatially adjacent Cys (CXC or CXXC), such as CRPCPC, CSGCNC, CQPCAC, CPC, and CGC.

### 2.3. Biophysical Characterization of CyDisCo Produced ECM Proteins

To be functionally active or to be suitable for structural studies, the proteins produced by CyDisCo should be folded i.e., not only quantity but also the quality of the protein produced should be evaluated. While non-reducing SDS-PAGE implied a single redox state and ESI-MS suggested that the purified proteins contained the expected disulfide bonds, they do not give information on the folding state.

The secondary structure of the purified proteins was examined by circular dichroism (CD). Based on the recently solved crystal structure of a VDF-D domain from von Willebrand Factor (pdb code; 6N29 [23]), the purified VWF-D domains of mucin 2 and alpha tectorin would be expected to be mainly comprised of beta sheets, with the remainder divided in approximately equal parts between alpha helices and random coils. Estimation of secondary structure from CD requires data to 190 nm or more preferably 180 nm. Due to maintenance of near-physiological levels of ionic strength, it was not possible to collect CD data below 197 nm. However, the CD spectra obtained for the purified VWF-D domains from mucin 2 and alpha tectorin were consistent with the expected secondary structure for the native protein (Figure 4A).

There are no structures available for any domain of region 3 of perlecan, but estimates can be made based on the three laminin IV type-A domains and eleven laminin EGF-like domains which together form 95% of region 3. Structural information for laminin EGF-like domains in other proteins exists (pdb codes 1KLO [45], 1NPE [46], 4AQS [47], 4 WNX [48] and 4URT [49]). Based on these, the 47% of region 3 formed from this domain type should be circa 20% β-sheet and 3% helices. Structural information for laminin IV type-A domains from other proteins (pdb 4YEP [50], 4AQS [47], 4WNX [48] and 4URT [49]) implies that the 48% of region 3 formed from this domain type should be circa 40% β-sheet and 10% helices. Hence, region 3 would be predicted to be circa 30% β-sheet and 3% helices, with the remaining 64% not being in regular secondary structure. The CD spectra of the purified perlecan constructs produced using CyDisCo are consistent with this (Figure 4B).

Folded extracellular proteins might be expected to be thermostable. The thermal stability of the folded state of the purified proteins was analyzed by thermofluor assays. Purified mucin 2 and alpha tectorin VWF-D domain-containing constructs did not exhibit visible denaturation even at 90 °C (Table 2), indicating that they are extremely thermostable. Similarly, all the constructs from region 3 of perlecan show a high thermal stability (Table 2 and Figure 5A). The two N-terminal constructs, G503-Q874 and S762-R1158, show a single-phase transition, indicating cooperative unfolding of domains in this region, with a T_m_ of over 80 °C. In contrast, the C-terminal region R1158-T1672 shows a two-phase transition, indicating uncooperative unfolding of the domains. The first thermal transition state occurs at 49 °C which can cause partial unfolding of the protein while the second occurs at 66 °C resulting in full unfolding of the protein. Full-length region 3 shows three transitions, consistent with those seen in the individual N- and C-terminal regions suggesting no cooperativity in unfolding between distal domains.

To confirm the high thermal stability of mucin 2 and alpha tectorin VWF-D domain-containing constructs thermal stability was also examined by CD. Both constructs showed temperature dependent changes in CD spectra consistent with a thermally induced conformational change. Fitting to the temperature dependence of the spectral changes indicated mid-points of 67 and 75 °C respectively for mucin 2 and alpha tectorin. However, both constructs retained substantial regular secondary structure even at 90 °C (Figure 5B,C).

Taken together the ESI-MS, CD and thermofluor data suggest that the proteins produced are correctly folded to their native state.

## 3. Materials and Methods

### 3.1. Bioinformatics

Domain boundaries were assigned based on sequence alignment, literature and secondary structure predictions. The sequence alignment was done using clustal omega [51], while Jpred [52] was used to make secondary structure predictions. Due to uncertainty in domain boundaries and the potential packing of domains in the full-length protein a number of constructs were made for each protein.

### 3.2. Cloning

Gene constructs for mouse perlecan and human alpha tectorin were ordered codon optimized for *E. coli* from GenScript and human mucin 2 gene containing plasmid was kindly provided by Gunnar C Hansson (Department of Medical Biochemistry, University of Gothenburg, Sweden). From these selected gene fragments were amplified by PCR. The gene fragments were cloned with restriction digestion and ligation into a modified pET23-based vector with a pTac promoter replacing the T7 promoter [18] using the restriction site pairs NdeI/EcoRI for perlecan, NdeI/HindIII for mucin 2 and NdeI/BamHI for alpha tectorin. In all cases this resulted in an *N*-terminal hexahistidine tag prior to the first amino acid of the protein sequence (MHHHHHHM-). The gene inserts in the plasmids made were fully sequenced prior to use.

Initial screening was performed on more than 140 constructs of perlecan, mucin and alpha tectorin, with the largest constructs giving rise to soluble expression being taken forward for further analysis or combined and rescreened. For example, for region 3 of Perlecan 37 constructs were initially screened (P520-E730, P520-G763, P520-P813, P520-Q874, D538-E730, D538-G763, D538-P813, D538-Q874, S762-Q874, S762-Q1125, P813-Q1125, P813-R1158, K923-Q1125, K923-R1158, S939-Q1125, S939-Q1158, R1158-G1267, R1158-P1324, R1158-L1562, R1158-P1612, Q1207-P1234, Q1207-E1529, Q1207-L1562, Q1207-P1612, Q1207-T1672, G1274-F1529, G1274-L1562, G1274-P1612, G1274-T1672, P1324-E1529, P1324-L1562, P1324-P1612, P1324-T1672, Q1342-E1529, Q1342-P1612, Q1342-T1672 and E1561-T1672). Based on the expression results from these three longer constructs were made (G503-Q874, S762-R1158, R1158-T1672) which covered the whole of region 3. Based on the results from these a construct covering the whole of region 3 (G503-T1672) was made (Table 3).

### 3.3. Expression

Plasmids with the gene of interest together with the plasmid containing the CyDisCo components were co-transformed into chemically competent *E. coli* cells and spread onto Lysogeny Broth (LB) agar plates supplemented with 35 µg/mL of chloramphenicol and 100 µg/mL of ampicillin for selection. After overnight incubation at 37 °C these were used to inoculate 5 mL of LB media supplemented with 2 g/L of glucose, 35 µg/mL of chloramphenicol and 100 µg/mL of ampicillin. These starter cultures were grown overnight at 30 °C, 250 rpm (2.5 cm radius of gyration) and were used to seed the cultures in 1:100 ratio.

Expression tests to screen for optimal conditions were carried out for the constructs in 24 deep well plates. The constructs were co-expressed together with CyDisCo components from pMJS205 [18] in up to four different *E. coli* strains (BL21(DE3) (from Stratagene), the Keio collection parental strain BW25113 [53], MDS42 [54] and BL21 (DE3) Δ*trxA/trxC* [55]), up to two different media (chemically defined [56] and terrific broth autoinduction media (Formedium) with trace elements and 0.8% glycerol and at two different temperatures (30 °C and 15 °C). All the constructs were found to be best expressed at 15 °C, in BL21(DE3) cells.

Main cultures were grown in 1 L flasks with 100 mL culture in each flask. The flasks were covered with oxygen permeable AirOtop (Thomson) membranes to ensure proper oxygenation of the cultures and first incubated at 30 °C, 250 rpm. Full length region 3 of perlecan was produced in both autoinduction media, while mucin 2 and alpha tectorin were produced in terrific broth autoinduction media and other constructs were produced in chemically defined autoinduction media. As all the constructs were expressed in auto-inducing BL21(DE3) cells, when the OD_600_ of the cultures reached approximately 4 - 6, the temperature was changed to 15 °C. Cultures were then incubated 18 - 20 h at 15 °C, 250 rpm. The cells were collected by centrifugation at 6,500× *g*, 4 °C. The supernatant was discarded, and the cells were frozen at −20 °C.

### 3.4. Protein Purification

Cells were re-suspended in the lysis buffer (20 mM phosphate buffer pH 7.4, 150 mM NaCl, 15 mM imidazole and 20 μg/mL DNase) and sonicated for 90 secs with, 5 secs pulse on and 25 secs pulse off at 40% amplitude. After centrifugation at 30,500× *g*, 4 °C for 40 min, the supernatant was collected and filtered through a 0.45 µm filter.

The large-scale purification of the protein constructs was carried out in a three-step process: immobilized metal affinity chromatography (IMAC), ion exchange and size exclusion chromatography (SEC).

The initial purification step was carried out using nickel IMAC. Nickel was loaded onto a HiTrap™ 5 mL chelating HP column (GE Healthcare). The column was then washed with 10 column volumes (CV) of millipore water and equilibrated with equilibration buffer containing 20 mM phosphate buffer pH 7.4, 150 mM NaCl. The soluble fraction was then loaded onto the column followed by 3 CV equilibration buffer. 10 CV of 20 mM phosphate buffer pH 7.4, 300 mM NaCl, 50 mM imidazole was used as wash buffer. After the wash, 3 CV equilibration buffer was run through the column. The protein was eluted by applying a linear gradient of 20 mM phosphate buffer pH 7.4, 150 mM NaCl, 300 mM imidazole over 10 CV.

Anion Exchange (AnEx) was used as the next step of the purification. Resource Q™ column (GE Healthcare) was washed with 10 CV of millipore water and then equilibrated with 20 mM phosphate buffer pH 7.4 of equilibration buffer. The eluted fractions from IMAC that contained the target protein were pooled and diluted 10x with equilibration buffer and loaded onto the column followed by 3 CV of equilibration buffer. Bound proteins were eluted by applying a linear gradient over 10 CV of 20 mM phosphate buffer pH 7.4, 500 mM NaCl for mucin 2 and perlecan constructs, and 20 mM phosphate buffer pH 7.4, 1 M NaCl for alpha tectorin.

HiLoad™ 16/600 Superdex™ 75pg column (GE Healthcare) was used for purification of mucin 2 and alpha tectorin constructs and HiLoad™ 16/600 Superdex™ 200pg column (GE Healthcare) was used for perlecan constructs. The column was first washed with 1 CV of millipore water and then equilibrated using 20 mM buffer containing 150 mM NaCl at a flow rate 1 mL/min. Different buffers were used for individual constructs; phosphate buffer (pH 7.4) for mucin 2, tris buffer (pH 8.0) for alpha tectorin and HEPES buffer (pH 7.5) for the perlecan constructs. These buffers were chosen after initial screening for optimal conditions for crystallization for individual proteins based on dynamic light scattering (DLS) and thermofluor analysis. The eluted fractions from anion exchange chromatography containing the target protein were pooled and concentrated to approximately 1 - 1.5 mL. This was then injected into the equilibrated column and the sample was eluted using corresponding buffer for each construct.

Reduced and NEM trapped, non-reduced samples of the fractions from all purification steps were analysed on 15% SDS-PAGE gels. For NEM-trapped samples a 1:8 mix of a stock solution of 250 mM NEM and sample were incubated for 10 min at room temperature before the addition of SDS sample buffer.

### 3.5. Biophysical Characterization

Far-ultraviolet circular dichroism (CD) spectra of the purified protein constructs were measured using a Chirascan-Plus spectrophotometer. All the scans were measured in duplicate at 22 °C as an average of 4 scans using a cell with a path length of 0.1 cm, scan speed of 1 nm/s, step size of 1 nm, and a spectral bandwidth of 1 nm. The wavelength range was between 190–250 nm. Data where the HT voltage value was above 750 V was excluded from analysis. The final spectrum was averaged and the blank subtracted. The final concentration of protein sample was 0.07 - 0.1 mg/mL in 20 mM phosphate buffer pH 7.4, 150 mM NaCl. The data was analyzed using Igor software (WaveMetrics Inc.). For thermal dependence, the proteins were diluted into water. At each temperature, individual scans were collected at a scan speed of 1 nm/s, step size of 1 nm, and a spectral bandwidth of 1 nm. The wavelength range was between 190–250 nm. There was a 2 °C step between spectra at 1 °C/min ramp rate and data where the HT voltage was above 750V was excluded from analysis. The mid-point for temperature dependency of the CD spectra was calculated using Global 3 thermodynamic analysis software (Applied Photophysics).

Thermofluor assay was performed using a 7500 Real Time PCR system (Applied Biosystems). The thermal cycling went from 20 °C to 90 °C with 2 °C interval per minute, and the fluorescence signal measured. For the analysis, 22.5 µL of 0.5 mg/mL protein sample in 20 mM phosphate buffer pH 7.4, 150 mM NaCl, was mixed with 2.5 µL of 50x of SYPRO-Orange protein cell strain (Sigma) dye. The final reaction mixture was 25 μL per well, having 4 replicates per sample. The mixture was transferred to MicroAmp^®^ Optical 96-well reaction plate (Applied Biosystems) and the plate was sealed using adhesive film (VWR). The melting temperature (T_m_) was determined from the derivative of the fluorescence signal using Igor software (WaveMetrics Inc.).

The molecular weights of purified protein samples were measured by electrospray ionization mass spectrometry combined with liquid chromatography (LC-ESI-MS) using a Synapt G2 HDMS Q-Tof (Waters) spectrometer or Q Exactive Plus Mass Spectrometer. For non-reducing conditions, 0.5 mg/mL protein samples were mixed with 0.1% trifluoroacetic acid (TFA) prior to analysis. For NEM-trapped samples the protein was incubated with a 1/8^th^ volume of 250 mM NEM for 10 min and the reaction was stopped with 0.1% TFA prior to analysis.

## 4. Conclusions

The results presented in this study provide a strong proof of concept of the use of the CyDisCo system for the production of complex disulfide-bonded proteins in the cytoplasm of *E. coli*. The final purified proteins appear to be in a folded state with all cysteines forming disulfide bonds. Though the yields obtained are not high enough for industrial scale production, they are sufficient for biophysical, functional, and structural characterization of proteins. The production of correctly folded protein with 44 disulfide bonds at purified yields of 4.4 mg/L greatly extends the potential use of *E. coli* for the production of eukaryotic proteins involved in a myriad of cellular processes.

## Figures and Tables

**Figure 1 ijms-21-00688-f001:**
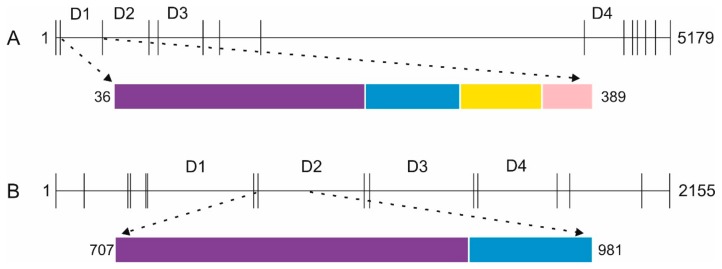
Schematic of the domain organization of mucin 2 and alpha tectorin. (**A**) Human mucin 2, (**B**) human alpha tectorin. Both are large multi-domain proteins. Amino acid numbering for the proteins and the domain constructs are shown. Different domains are colored separately: von Willebrand Factor D (VWF-D) domain (purple), C-8 region (blue), TIL domain (yellow), E-8 region (pink). The D-assemblies are represented by “D” [23]. Vertical lines donate domain boundaries. For clarity other domain types are not shown.

**Figure 2 ijms-21-00688-f002:**
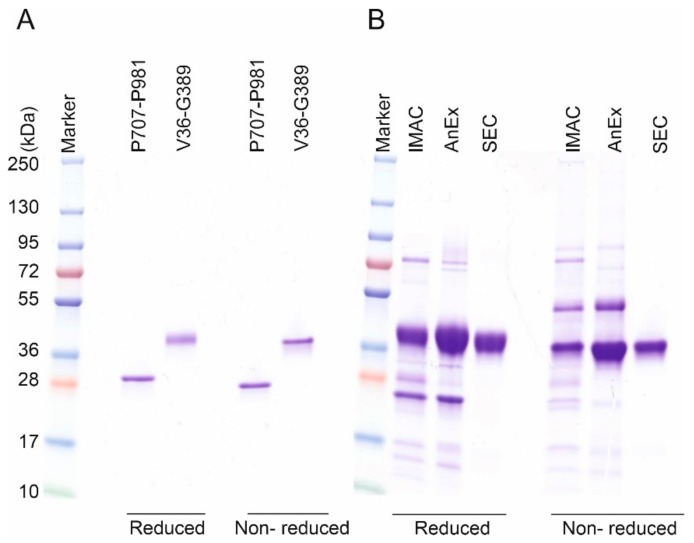
Representative coomassie stained SDS-PAGE gels. (**A**) Reduced and non-reduced samples of purified alpha tectorin (P707-P981) and mucin 2 construct (V36-G389). (**B**) SDS-PAGE gel of the mucin 2 construct after each purification step. An additional slower moving species can be seen in the non-reduced NEM treated samples after IMAC and anion exchange (AnEx), suggesting at least one additional redox state, which may involve a disulfide-linked hetero- or homo-dimeric state degradation product.

**Figure 3 ijms-21-00688-f003:**
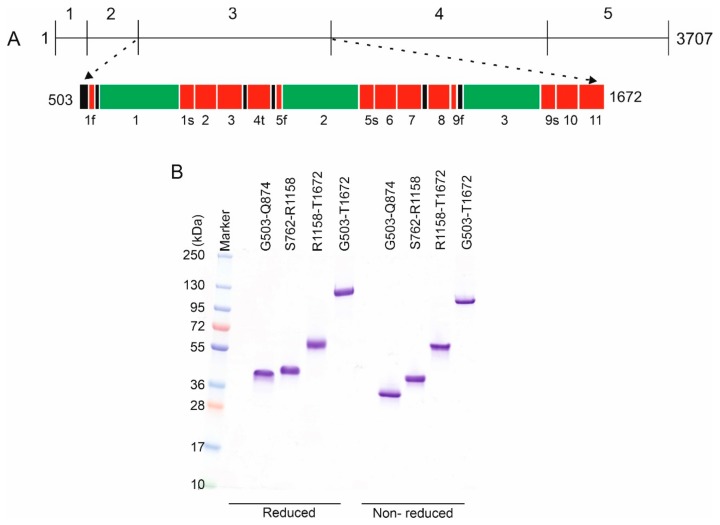
Production of region 3 of perlecan. (**A**) Schematic of the domain organization of perlecan. Amino acid numbering for the proteins and the domain constructs are shown. Different domains are colored separately: laminin IV type A-like domain (green), laminin Epidermal Growth Factor (EGF)-like domain (red) and unassigned sequence regions (black). Numbering (1–5) shows different regions of perlecan [30]. Region 3 contains 3 laminin IV type A-like domains and 11 laminin EGF-like domains. The laminin IV type A-like domains split the laminin EGF-like domains 1, 5 and 9 into two parts (where “f” stands for first part, “s” for second part, “t” stands for truncated). (**B**) Representative coomassie stained SDS-PAGE gels of reduced and non-reduced samples of purified region 3 constructs of perlecan. Only single bands are visible in the non-reduced samples suggesting that the proteins are in a single redox state.

**Figure 4 ijms-21-00688-f004:**
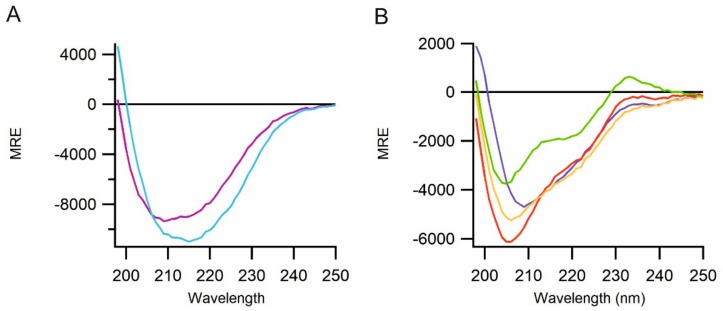
Far-ultraviolet circular dichroism (CD) spectra for the purified proteins. (**A**) The spectra of mucin 2 (V36-G389; light blue) and alpha tectorin (P707-P981; purple). (**B**) Spectra of constructs of region 3 of perlecan; G503-Q874 (dark blue), S762-R1158 (red), R1158-T1672 (green) and G503-T1672 (orange). All constructs show spectra consistent with their expected natively folded state at near-physiological ionic strength.

**Figure 5 ijms-21-00688-f005:**
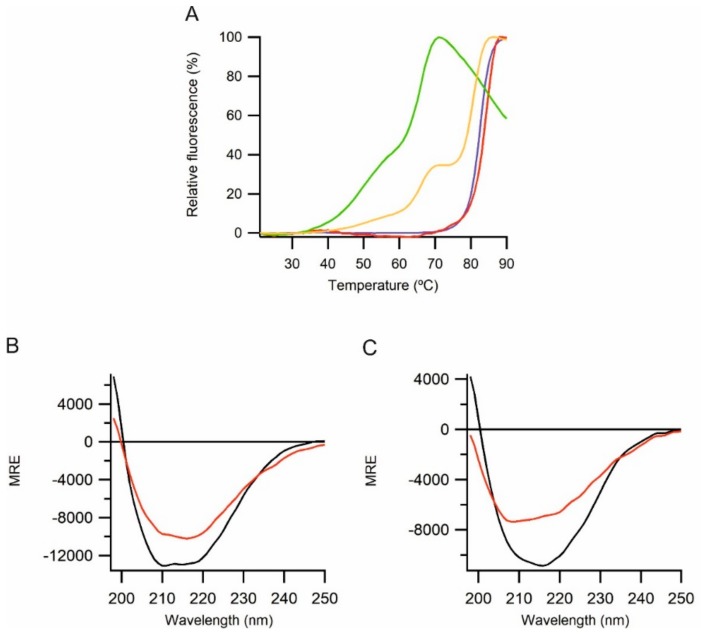
Thermal stability of the purified protein. (**A**) Thermal stability of perlecan region 3 constructs based on thermofluor analysis. A single step thermal transition can be seen for G503-Q874 (blue) and S762-R1158 (red), and a multi-step thermal transition for R1158-T1672 (green) and G503-T1672 (orange). (**B**) and (**C**) Thermal stability of mucin 2 (**B**) and alpha tectorin (**C**) VWF D-domain-containing proteins. CD spectra at 22 °C (black) and 90 °C (red).

**Table 1 ijms-21-00688-t001:** Molecular weight analysis by electrospray ionization mass spectrometry (ESI-MS) for purified proteins. The theoretical average molecular weight (M_theor_) of the His-tag perlecan constructs in dalton (Da) were calculated using ExPaSy ProtParam tool [41]. The experimental molecular weight (M_exp_) was determined by mass spectrometry. Formation of one disulfide bond reduces the calculated theoretical molecular weight by approximately 2 Da. The same masses were obtained with NEM-treatment. The results suggest all cysteines are in disulfide bonds in the perlecan constructs analysed.

Construct	# of Cysteine	M_theor_ (Da)	M_exp_ (Da)	Δ mass
Perlecan MH6M-(G503-Q874)	26	41713	41688	25
Perlecan MH6M-(S762-R1158)	30	43712	43681	31
Perlecan MH6M-(R1158-T1672)	48	56079	56030	49
Perlecan MH6M-(G503-T1672)	88	127264	127173	91

**Table 2 ijms-21-00688-t002:** Thermofluor analysis to determine the melting temperature of the purified proteins. A single-phase transition is observed for perlecan G503-Q874 and S762-R1158. A two-phase transition is observed for R1158-T1672. A three-phase transition can be seen for the full-length region 3 (G503-T1672). Mucin 2 and alpha tectorin denaturation was not observed even at 90 °C.

Construct	T_m_ (°C)
Mucin 2 (V36-G389)	Above 90
Alpha tectorin (P707-P981)	Above 90
Perlecan (G503-Q874)	83
Perlecan (S762-R1158)	84
Perlecan (R1158-T1672)	49, 66
Perlecan (G503-T1672)	51, 66, 81

**Table 3 ijms-21-00688-t003:** Plasmids used in this study.

Plasmid	Construct Details	Reference
pMG01	MH_6_M-human mucin (V36-G389)	This study
pMG03	MH_6_M-human alpha tectorin (P707-P981)	This study
pAS20	MH_6_M-mouse perlecan (G503-Q874)	This study
pAS04	MH_6_M-mouse perlecan (S762-R1158)	This study
pAS05	MH_6_M-mouse perlecan (R1158-T1672)	This study
pAS39	MH_6_M-mouse perlecan (G503-T1672)	This study
pMJS205	CyDisCo: Erv1p and PDI	[18]

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
