# Peer review of "Production of Extracellular Matrix Proteins in the Cytoplasm of E. coli: Making Giants in Tiny Factories"

_ijms, 2020, doi:10.3390/ijms21030688_

Round 1

Reviewer 1 Report

In this manuscript the authors describe the production of mammalian extracellular matrix proteins containing between 8 and 44 disulfide bonds in the cytoplasm E. coli. To achieve this they overexpress two eukaryotic proteins responsible for the formation and isomerization of disulphide bonds: Ero1 and Erv1p. This system already been described elsewhere but was for the first time successfully applied to protein containing more than 5 disulfide bonds.

What I am missing in the manuscript is a more detailed description in the Results section of the optimization of the expression conditions. As it is now the authors only summarized the conditions (strains, media and temperature) in the Materials and Methods section and state under which conditions the best results were obtained.

Furthermore, a more detailed description of the CyDisCo process would be very helpful.

Other comments:

In sentence 146-148 the authors state that ‘during subsequent purification the natively folded protein was isolated’. Please add the nature of this additional purification.
Details or a reference is missing is missing for plasmid MJS205.

Please give the details of the gradients used in the different purification steps.

Please give the concentration of the buffers used for size exclusion chromatography.

The last two paragraphs of the Material and Methods section can be deleted.

Reviewer 2 Report

RE: “Production of extracellular matrix proteins in the cytoplasm of E. coli: Making giants in tiny factories”

Comments to the authors

In the manuscript “Production of extracellular matrix proteins in the cytoplasm of E. coli: Making giants in tiny factories” Sohail and co-workers explored the limitation of the previously developed E. coli-based protein production system CyDisCo. First, they produced three protein domains originating from eukaryotic extra cellular matrix proteins, with different numbers of disulfide bonds. Then, they purified and somewhat characterized the folding state of the proteins produced.

This manuscript presents interesting data that is relevant for scientist working in the protein production field especially with the focus of producing eukaryotic proteins in a bacterial production system. Notably, producing these three disulfide bond rich proteins is an important achievement. However, this reviewer has major concerns about the experimental setup and the data interpretation of the biophysical characterization of the produced proteins.

Main concerns:

The authors compare reduced and non-reduced, NEM treated samples to monitor if free cysteine residues are present in their produced proteins. Unfortunately, positive controls are missing to verify that NEM can indeed be linked to their proteins. Without such controls it is impossible to be sure if actually all cysteine residues have formed disulfide bonds.

This reviewer suggests that Figure 1 C and 2 C are combined and integrated into section 2.2, so that the data regarding the perlecan is linked to the part describing this data.

The authors use CD spectroscopy to assess the folding of their produced proteins. Although this method is well suited for this purpose, the authors did not record the full spectra starting from 180 nm. However, to analyze the secondary structure of any protein the spectrum between 180-200 nm is of high importance and without this information structural prediction cannot be made accurately. Moreover, the authors concluded that the produced mucin and alpha tectorin have the expected regular secondary structure based on comparisons with crystal structure. To make such a statement, global curve fitting is needed to calculate the amount of alpha-helical, beta sheets, beta turn and random coil structures. Only then a semi-quantitative comparison between crystal structure data and the here recorded CD data is appropriate.

Similar comments apply to the CD data regarding the perlecan domains. Additionally, these spectra (Figure 3 B) indicate a significant amount of random coil structures as indicated by the local minima at 205 nm. Therefore, it seems unlikely that the produced perlecan domains “have the expected regular secondary structure” since the referenced crystal structures show mainly beta sheets and beta turns but no unstructured regions.

Finally, the authors determine the thermosstability of the produced proteins using the thermofluor assay. In this assay mucin 2 and alpha tectorin exhibited high melting temperatures above 90°C. An alternative explanation for this high Tm is that the dye does not bind to these to proteins even if they are in an unfolded state. To exclude this possibility this reviewer strongly suggests to use CD spectroscopy monitoring the signal at 220 nm while the sample is heated up. If the CD signal is indeed stable at elevated temperatures a high Tm can be assumed.

Minor points:

line 31: “In case of prokaryotes this occurs…” should be changed “In case of Gram negative bacteria this occurs…”

line 44: “…has a lower capacity” capacity of what?

line 63: “…that all six were homogeneously folded” This statement is to strong, since CD spectroscopy does only give an average signal for all folding states present in a sample. Please rephrase by something like: “…all six proteins were present in a folded state”

line 76: “many lack other essential PTMs.” Please add reference. To the best of my knowledge at least mucins are highly glycosylated.

line 86: please add corresponding references

line 92: please add an appropriate reference

line 104: Figure 1 C. “4t” is not explained in the figure legend, does “t” stands for total?

line 114: “Since E. coli makes proteins >100kDa in size in low yields…” please add  references supporting this statement

line 133: Figure 2 B. It would be helpful for the reader if the bands that are described in the figure legend are labeled in the figure.

line 138: “hetero-dimeric state” To this reviewer it seems likely that the degradation product (~25 kDa) forms a homo-dimer resulting in the higher band at ~50 kDa. Please also consider this possibility in lines 145-151.

line 151: “c.f.” should be replaced by “compared to”

line 159: “Mutations in region 3 of perlecan are linked to…” Does this involve mutations in cysteine residues?

line 197: “…all cysteines were in disulfide bonds” A positive control is required to verify that the NEM-treatment would work on free cysteines in these proteins. Please reduce the samples and then treat them with NEM.

line 215-218: Please elaborate on this. Which secondary structures were observed in the referred crystal structures? Which signals in the CD curves correspond to the respective secondary structures?

line 227: “…both would be expected to be thermostable.” Please explain this or add an appropriate reference.

line 263, 289, 291: one space to much

line 305: “30500” should be replaced by “30,500”

line 329: Why were different buffers used for the different constructs?

line 360-371 should be removed

line 374-375: “…appears to be homogenous and correctly folded in their native state.” It was neither shown in this manuscript that the produced proteins were homogenously folded neither was any biological activity demonstrated. Therefore, this statement should be changed to “The final purified proteins appear to be in a folded state with all cysteines forming disulfide bonds.” or similar.

Reviewer 3 Report

In this study, Sohail et al explore the ability of E. coli engineered with a system previously known as CyDisCo (cytoplasmic disulfide bond formation in E. coli) to produce properly folded eukaryotic proteins with a high number (8 to 44) of disulfide bonds. The topic of this study is important and valuable, and the manuscript is clear and well written. However, I have some concerns regarding the interpretation of the data. In my opinion, further pieces of evidence are needed to support the conclusions of the manuscript. I would recommend to reconsider the MS after major revision.

The title of the manuscript is not fully accurate considering the content of the study. Authors should refer to “extracellular matrix protein domains 2: I am a bit skeptical regarding such little shift in migration when comparing non-reduced and reduced conditions. If there were so many disulfide bonds in the protein domains, I would rather expect a significant shift in migration when comparing these conditions. I only see a substantial shift for G503-Q874 (Fig. 2C). No shift in R1158-T1672 (Fig. 2C), and very small shift elsewhere. Page 6, line 207 and Table 1. The statement “suggested that the purified proteins contained the expected disulfide bonds” is, in my opinion, overstated. Analysis by electrospray ionization mass spectrometry suggests that all the cysteines are engaged in disulfide bonds. Although the authors claim that the cys-rich proteins expressed are homogeneous and well folded, I think that full evidence is missing. Mass spectrometry could be performed after proteolysis digestion in order to analyze and determine which disulfide bonds are made (ex: cys 36 with cys 159; cys 80 with cys 528 etc…). This strategy might not be able to resolve all the protein sequence, but it should identify a substantial number of disulfide bonds. It will also show whether the population is homogenous and whether the results are in agreement with the literature regarding the known disulfide bounds. Page 6, line 195: It would have been good to have a positive control protein that has some cys reacting with NEM. Page 7, line 228 and Table 2. The fact that some constructs have such high denaturation temperature (>80 °C) make them excellent candidates for crystallization. This might be another possible alternative to prove that the expected bonds are made in these proteins (in case proteolysis/mass spec does not work).

Minor comments:

Page 1, line 18. Correct “,.” Page 1, line 24. No space between extra and cellular Page 1, line 29. “For around one-third of human proteins folding to the native state includes disulfide bond formation”. Please include a ref Page 2, line 44. “while the periplasm has a lower capacity”. What do you mean? Page 2, line 46. “CyDisCo (cytoplasmic disulfide bond formation in coli)” You refer to ref 12-15 here? Page 3, line 93. “It also acts as a sieve that only lets in nutrients and keeps harmful toxins out”. How is it possible? Say more or give ref Page 3, line 94. “get to” should be replaced with “reach” 2B. I am surprised you could get rid of the high MW species with the SEC Page 5, line 138. What make you think that the disulfide bound involve a degradation product ? Page 5, line 176. If I understood correctly, perlecan is a membrane protein or a membrane-associated protein? It is then normal that the 3 engineered domains are soluble? Page 8, line 239. Please correct with “cooperative” Page 8, line 272. I think pET23 has a T7 promoter, not a ptac one?

Round 2

Reviewer 2 Report

In the revised version of the manuscript “Production of extracellular matrix proteins in the cytoplasm of E. coli: Making giants in tiny factories” Sohail and co-workers provide appropriate response to my previous comments.

Reviewer 3 Report

Point 1: ok to leave the title as it is, since the abstract specifies “domain constructs”.

Point 2: I will leave the benefit of the doubt because, as a reviewer, I cannot predict or decide where your bands are supposed to migrate! However, I must say that the figure provided in the rebuttal letter did not change my mind. In this example, a nice shift is visible for the WT that only contains 3 disulfide bonds. Mutants lacking one disulfide bond do not migrate to the same position as the wild-type. The authors hypothesize that disulfide bonds can have opposite (faster or slower) effects on migration and the occurrence of multiple disulfide bonds can have compensatory effects on migration. It can certainly be true in some cases, but proteins with > 15 disulfide bonds will probably have some of them shifting the migration in one way or another.  Data are convincing for the G503-Q874 and G503-T1672 constructs, but I’m still concerned about the others. In my opinion, another alternative and more convincing experiment would have been to perform the thermofluor analysis on both the non-reduced and reduced forms. This should show a huge Tm difference between them.

Point 3: What I meant in my previous comment is that the whole point of the MS is to prove that the produced proteins are folded and contain the appropriate disulfide bonds. In some places of the revised MS, the term “suggest” is used, but in others for example it is said “analyses indicated that the purified proteins are most likely homogeneously correctly folded into their native state”, “Hence, they are probably natively folded”. The other thing that was misleading in your previous sentence is the use of the word “expected” in “expected disulfide bonds”, since in the revised version you admitted that “the native disulfides have not been previously mapped, they are only presumed by similarity with proteins with homologous domains”.

Moreover, I did not specifically ask for disulfide mapping for a given construct (I agree that perlecan would be difficult) and I did not ask to map all the disulfides within a protein. It could be attempted for tectorin or mucin protein, which contain much less disulfide bonds (8 and 15, respectively). I regret that the authors did not even try to do such experiment. That would have convinced me and, more importantly for the authors, strengthen the impact of the study. At the very least the authors need to tune down their “most likely” and “probably” statements.

Point 4: it is good that the authors added the requested control experiment.

Minor point 1: sorry I was not clear enough and you can just ignore my comment. I just meant that the two products were fairly close in size (40 and 50 kDa), which may be a bit difficult to fully separate as non-overlapping peaks in GF.